# When Is Choice Empowering? Examining Gender Differences in Varietal Adoption through Case Studies from Sub-Saharan Africa

**Vivian Polar** [1],*, **Jaqueline A. Ashby** [2], **Graham Thiele** [1] **and Hale Tufan** [3]

1   Research Program on Roots, Tubers and Bananas (RTB), Led by CIP, Consultative Group for International Agricultural Research (CGIAR), Lima 12, Peru; rtb@cgiar.org
2   Independent Consultant, Norwich NR1 4PT, UK; jacqueline.ashby@cantab.net
3   Department of Global Development, Cornell University, New York, NY 14853-0001, USA; hat36@cornell.edu
*   Correspondence: v.polar@cgiar.org; Tel.: +51-952-475-348

**Abstract:** This paper examines the question of what makes choice empowering and critiques prevalent approaches to empowerment focused narrowly on agency as the ability of individuals to make their own free choices and act independently. The implications of a narrow focus on agency are illustrated with the examples of technology choice in agriculture, specifically choices involved in the adoption of improved plant varieties. This example elucidates the limits of individual agency and permits an analysis of how choices may be structured to be either empowering or disempowering, with examples from specific plant breeding cases. In view of the importance given to equitable choice of technology for closing the gender gap in agricultural productivity and sustainability, the paper explores what practical steps can be taken towards a balanced approach to empowerment. An approach to designing a new plant variety by constructing choice differently is illustrated, using information on gender relations. The paper derives lessons from the plant breeding cases to inform other kinds of interventions, so that work on how choices are defined is given as much importance for empowerment as creating the option to choose. Agents who exercise power over rules and resources can either reproduce the status quo or innovate; thus, a balanced approach to empowerment requires careful analysis of the elements of choice.

**Keywords:** empowerment; choice; agency; gender; new varieties; improved varieties; plant breeding

## 1. Introduction

The concept of women's empowerment has become pivotal to international development practice. Empowerment of women is a sustainable development goal, defined as a process that results in the expansion of women's ability to make choices affecting their lives and situations [1]. For developing countries, empowerment of women is strongly associated with progress out of poverty [2,3]. To this end, a growing diversity of development interventions include women's empowerment as a program objective and are designed to enhance women's empowerment proactively [4–7]. However, the record of actual empowerment as an outcome is uneven and it remains unclear what empowerment means or how to achieve it [8,9].

This paper argues that one reason for the spotty record of women's empowerment in development efforts relates to how the notion of choice is defined. The current focus in development is on empowerment achieved through an increase in women's agency, defined as an ability to set goals, exercise meaningful choice and achieve desired outcomes, giving them more say in decision-making, enabling development of their capacities and producing, as a result, better access to key productive resources and opportunities [10–13]. For example, assessing progress in women's empowerment in developing countries is heavily oriented towards measuring changes in agency in decision-making, defined as the capacity of individuals to make their own free choices and act independently [14], such as in household spending decisions [15]. While the existence of structural impediments to

choice-making, such as norms and institutions, are given token recognition, the focus of development interventions are on agent empowerment—on individual women's capacities, actions and achievements in decision-making over productive assets, income and household relations [16,17]. Strengthening these dimensions of empowerment can have important benefits for the welfare of resource poor women and their families, for example, by improving productivity [18]. However, the focus on agency as the exercise of choice means there is an imbalance in the approach that has significant implications for policy and development practice. A focus on agent empowerment—what Haslanger calls the individualistic approach to oppression [19]—is an invitation to ignore the need for change in structures as well as in individuals. Along the same lines, research analyzing the choices and decisions of individual women from Kenyan, Tanzanian and Syrian rural communities shows that these choices and decisions depend on the empowerment of significant others and their "collusion" in women's and men's deviations from prevailing gender norms; it also demonstrates how structural influences on choice that are beyond the individual's control, play out through interpersonal relations [20]. This dynamic suggests there are important preconditions for empowerment rooted in the ways that choices are defined by institutions and through interpersonal interactions, which require deeper attention.

This paper makes a theoretical and practical contribution to understanding relationships between power, gender and technology by exploring the structure of choice. We examine what makes choice meaningful when structural impediments exist beyond the individual decision-maker's control, and how changes in the structure of choice can be understood as a precondition for empowerment. We illustrate the implications of a disregard for meaningful choice and the kinds of practical steps that can be taken to increase capacity to make meaningful choices through the case of technology choice in agriculture, specifically with reference to new crop varieties produced by improved plant breeding. The paper aims to synthesize key arguments and is not intended to be a comprehensive review of cases or survey research on variety adoption, particularly since most adoption studies are not designed with gender analysis in mind [21,22]. However, technology choice in agriculture is an excellent window for examining the structure of choice as a precondition for empowerment for two reasons.

The main reason is that analysis of technology choice calls attention to the ways agency and empowerment depend on the structure of choice beyond the individual's control, because an agricultural technology adoption decision involves two major determinants: first, in the domain of individual agency, is the adopter's resources and preferences; second, in the domain of structure, are the inherent attributes of the technology and the innovation system. Analysis of technology choice requires us to consider not only the capacity and preferences of the individual related to their agency but also the parameters of choice offered to the adopter that are decided by different actors elsewhere. The technology's inherent attributes in the case of an improved plant variety are the traits or characteristics that breeders decide to incorporate in a new variety. For breeders, a trait is a distinguishing, genetically and environmentally determined feature or quality of the variety that can be measured and prioritized as a target for breeding (e.g., growth habit, level of resistance to a disease, shelf-life, yield, starch, gluten or protein content). When people express varietal preferences, they often refer to characteristics of the crop, some of which are known traits, others of which require research to discover whether the preferred characteristic is associated with a heritable trait.

The second reason technology choice provides a fruitful example for examining the structure of choice is the history of low adoption of improved varieties, some of which have encountered resistance from women producers or have had detrimental outcomes for women [23–26]. Empowering women producers with access to the same choice of technology as male producers is seen as a vital component for closing the gender gap in agricultural productivity [2,27]. Agricultural innovation is expected to improve women's agency in gender relations but is heavily focused on individualistic agent empowerment in the household, value chain, or community context [28]. Given pervasive low yields in devel-

oping countries, new crop varieties are an essential component of productivity-enhancing technology and very frequently included in technology packages widely promoted for sustainable intensification or climate-smart agriculture. Hence, development-oriented plant breeding is expected to address gender equity through women's empowerment [29,30], and so provides a conducive context for a critique of the current approach to empowerment in development.

The paper is organized as follows. First, we examine the question of what makes choice empowering, with reference to technology choice in agriculture and gender differences in the adoption of improved plant varieties. Next, we examine how development-oriented plant breeding decisions may exercise power-over varietal choices for resource-poor men and women producers, with examples from specific plant breeding cases. We illustrate an approach to designing a new variety by constructing choice differently, using information on gender relations. We then discuss how the plant breeding cases can be utilized to argue that how choices are defined is as important for empowerment as creating the option to choose.

## 2. Power to Choose

In this section, we examine the question of what makes choice meaningful in the exercise of agency, and why meaningful choice is a precondition for empowerment. Any attempt to understand what leads to empowerment must take into account the different manifestations of power. In gender theory, the concepts of "power-over" and "power-to" structure much of the discussion. Power is viewed as domination that reproduces oppression, patriarchy and subjection [31–35]. It reflects a relationship that is unjust and oppressive to those over whom power is exercised. Feminist and other social movements have explored in depth the power-to dimension identifying two additional components: power-with [32,36], and power-from within [14,32], reflecting differences between power exercised collectively and internally, as a social psychological form of empowerment. The focus here is on contrasting what "power-over" and "power-to" mean for empowerment, agency and choice.

"Power-over" is defined as power to control others. This definition of power-over can also be related to both individual and collective spheres of life, where power-over may manifest through formal and informal social structures [37], related to institutions, their governing rules and the shared cultural norms these enforce, often interpreted as constraints on individual agency [38–40]. On the other hand, "power-to" is defined as the ability to make choices and the capacity or competence to transform agency into effective action through choice-making [14,32,35]. Choice is an outcome of individual agency. Social outcomes are ascribed to the choices of individuals, thus low adopters of modern varieties, like the poor or victims of natural disasters, are "bad choosers."

Different definitions of how power is manifested share a pivotal concern with agency in terms of the ability to make choices, where empowerment refers to the process by which the ability to make choices is acquired by those who have been denied that ability [41]. However, the power to make choices depends not only on individual agency [41], but also on collective social structures [42]. The concept of "empowerment through" refers to the way in which the exercise of agency is shaped by processes that are not within the individual's personal control but are the outcome of interpersonal relations that lend structure to individual choices [20]. The mutual interdependence of structure and agency, expressed as the process of structuration, means that structure can be enabling as well as constraining [39].

The importance of structuration is to expand analysis of the structure of choice beyond the individual "choice-taker" to consider the agency of other actors in defining that structure. These actors are "choice-makers" who whether deliberately or unconsciously, define the parameters of choice for the choice-taker. In this expanded analysis, the power, intentions and world-view of choice-makers are as important as the capacities and agency of choice-takers. This dynamic view of structure and agency requires development in-

terventions that address the need for change in how choices are structured as well as improvement in the capacity of women to make choices.

We identify two types of choice, differentiated by the conditions in which the power to choose is exercised: meaningful choice and restricted choice. For choices to be real, they must exist and be perceived as existent [41]. Meaningful choice must involve the existence of choice and the perception of choice among positive options that have value to the actor and are available to him or her; the result of the choice must be beneficial but may not be empowering. For example, exercise of the choice of attending a women's group, often used as an indicator of women's empowerment, is meaningful only if groups exist that women can choose to attend, and empowering only if women's ability to make choices improves their control over important resources or social processes as a result of group participation. If women do not have the means to create groups where none exist, then the choice is restricted [15]. It is evident that some parameters of important life choices are determined by institutions such as legal frameworks that are not in the domain of individual agency. An example is the terms of credit that determine whether or not a resource-poor woman actually benefits from the decision to take out a micro-credit loan. Although a woman may exercise power to decide by taking out a loan, the terms of some microcredit programs have made it possible for men in the household to control the capital and use it to their own benefit, with disempowering results for the women who must pay off the loan [43]. Restricted choice, therefore, involves not having positive options to choose from under disempowering conditions that prevent the actor from changing the rules of the game and generating better options. In the context of micro-credit, an empowering choice would involve both the power to take out a loan and a change in gender relations enabling women to sustain exclusive or dominant power over the use of the loan. In other words, empowerment is realized by the interplay of changes in individual "power to" choose and the structures they confront of "power over."

In this paper, we are particularly interested in understanding how restricted choice is manifested in technology design, and what steps can be taken to change this situation. Restricted choice can lead to the formulation of "inappropriate adaptive preferences," harmful to those who formulate them and conducive to self-subordinating choices [44,45]. An example of self-subordinating choice observed in Andean potato farming is women's preference for lower yielding native potatoes that are highly valued for family consumption but historically have had lower market value and require fewer inputs, especially labor [46]. In this instance, although women have the option of cultivating the more productive improved potato varieties, they have chosen to plant the less productive native varieties. In some instances, this can be a proactive choice, expressing a preference for the superior quality of less productive native potato varieties; however, in this case, it is a self-subordinating adaptation to a restricted choice, to accommodate their inferior access to labor, capital, farm equipment, seed, fertilizers, pest control products and market access [46].

Restricted choice may increase the options available for exercise of agency under disempowering conditions [44] but it can also produce outcomes for women that are contradictory, sometimes referred to as the "impact paradox" [47]. One example of the impact paradox is the situation where women gain the option of entering the labor market but, once employed, lack the power to bargain for decent wages and working conditions and may be worse off as a result. In West Africa, rural women work long hours peeling cassava in small-scale processing enterprises. Given the choice, they are resolutely opposed to mechanization of cassava peeling, even though this could greatly relieve their work burden, because they will not have any say over how mechanization restructures their workload or compensation [21]. In this situation, the women foresee they would be worse off choosing mechanization technology, when this is a restricted choice that does not include the added option of, for example, joining a cooperative that owns the machinery and shares the profits. The decision to enroll in microcredit is another example of the impact paradox: the microcredit loan potentially has tangible benefits for women, improving their

ability to generate cash income but it is a restricted choice because the choice does not include a way to avoid oppressive dependence on spouses who control the loans and the repayments [43,48,49]. In agriculture, there is a long history of studies reporting an impact paradox, in which women are relatively disempowered and materially worse off after the adoption of technological innovations that improve productivity [23,50–53]. A telling example is a study of dairy technology adoption in Ethiopia: overall household earnings from dairying increased substantially but with the result that women in technology adopter households worked significantly longer in dairying compared to women in non-adopter households. While the women adopters gained some income, their share of dairy income declined. The increased dairy income went disproportionately to men [25]. Each of these examples illustrates how women can gain options in the form of restricted choices that produce disempowering outcomes.

In summary, to understand how choice is structured, it is necessary to understand how the different kinds of power condition choice: that is, what is the capacity to exercise choice (power-to) and what is the structure of power-over the choice? Choice of technology in the case of plant breeding illustrates this juxtaposition of two kinds of power very neatly. On the one hand, producers make the choice about whether or not to adopt an improved crop variety. On the other hand, plant breeders control the set of options available for producers' choice, by deciding the set of desirable traits to be incorporated into a new variety. Breeders frequently consult producers about desired attributes of plant varieties, but by and large the decision on what to breed for still lies with the breeders, seed companies and agricultural bureaucracies. In so doing, breeders also decide what plant traits are *not* on offer to producers. In the next section, we examine what gender differences in the adoption of improved plant varieties tell us about meaningful choice versus restricted choice.

### 3. Meaningful Choice? Gender Differences in the Adoption of Improved Plant Varieties

Development-oriented plant breeding, such as the Green Revolution, launched to solve global food shortages, has usually prioritized increasing productivity of crops and the area devoted to improved varieties without paying much attention to demand from farmers or consumers [54]. Since the Green Revolution, studies of adoption of improved varieties show that women producers are less likely than men producers to adopt improved plant varieties [24,26,27,55–74].

Despite this body of literature, interpretation of the relationship between gender and adoption of varieties is not straightforward and gender is not always a consistent predictor of varietal choice [75,76]. A producer's behavior can seldom be explained successfully by a single identity such as his or her sex, which acquires social meaning in relation to other intersectional characteristics, including age, education, wealth, caste or ethnicity. Some studies that report no effect of sex of respondent on technology adoption include sex as a variable in multivariate analyses, but do not consider interactions between sex and other intersectional characteristics and are further limited by small sample size. The relationship between sex of the farmer and adoption can become insignificant in regression analyses that control for resources which affect adoption and are highly correlated with sex [77,78]. Some studies find no relationship between sex of respondent and adoption in their aggregate study sample, but with disaggregation they find that gender is a significant predictor for adoption in specific social groups, such as young women producers [26]. More problematic is that gender is frequently operationalized as the sex of the head of household, and so gender is confounded with household structure, ignoring the existence of women who farm and make adoption decisions as members of a male-headed household [79,80].

Recent studies that avoid confounding gender with household structure and analyze adoption decisions of individual men and women plot managers find that men tend to have higher rates of adoption of improved crop varieties, but there are exceptions, depending on the level of gender inequality [81,82]. One study found that lower-adopting women producers in Malawi have less access to seed of improved varieties than men, but in Zambia where women producers benefited from the commercialization and intensification

of farming, their level of adoption was the same as that of men [82]. In some cases, the factors explaining adoption by women producers are different from those that explain adoption by men [24,73]. This finding is logical when we consider that unequal power and resources mean that men and women engage in agriculture with different means of production and that, as a result, women have to develop different strategies for farming than men [83,84].

Although women producers tend to adopt improved crop varieties at a lower rate than men, the explanation for this difference remains far from clear [85–89]. Varietal trait preferences are recognized as important factors in adoption decisions [68,90–94], but it is rare for men's and women's preferences for different varietal traits to be included as a predictor in quantitative adoption studies, or even included in data collection survey questions, which remain a notable gap in adoption research [21,24,66,81,94]. Inattention to trait preferences in adoption studies is common because there is an assumption that improved varieties must be more attractive than existing varieties to all farmers, men and women alike, and this reflects part of a more generalized lack of attention to consumer/end-user preferences in public sector research [21]. Studies of gender differences in trait preferences are limited by reliance on simple comparisons of men and women, with no consideration of intersectional characteristics, and by widely different methodologies, which hampers generalization. In certain circumstances there is broad agreement among men and women producers about the importance of some desirable traits while they give different weights to another set of traits, many not considered as breeding objectives [30]. However, studies making simple comparisons of men and women have found different preferences for varietal attributes: some traits are exclusively important to women producers or are given more importance by women producers than by men. Frequently, modern plant breeding programs overlook quality traits considered indispensable for full adoption of a modern variety by women producers, but considered secondary to performance by breeders [24,60,63,68,83–85,95–101]. For example, women's varietal preferences often disfavor labor-increasing traits, such as "difficult to thresh" or "hard to peel" that increase the demand for unpaid female labor in the farm family [60,96,100,101]. Another example of gender-differentiation that reflects gender roles is the greater importance given in sub-Saharan Africa to cooking traits of banana by women (who do the cooking) compared with the importance given to beer-making quality traits by men (who sell it for beer making) [21,102,103].

Nonetheless, it is important to recognize that having a preference or an option is not the same as exercising a choice. For example, a study of improved drought-tolerant maize variety adoption in Uganda found that preferences do not explain women's lower adoption of new varieties, which was about half that on men's plots. This was because only 5% of the female spouses had decision-making power over the so-called women's plot they were cultivating so that most women have little opportunity to make an independent adoption decision [26]. In this case, formal structures of land tenure and gender roles and norms restricted women's choice despite the option of adopting new varieties.

Gender differences in varietal trait preferences are often associated with the notion that there are "women's crops," which, especially in Africa, is a term that farmers use to refer to crops usually grown by women and is, supposedly, preferred by women. The term "women's crops" does not describe in practice crops that are the monopoly of women and should not be understood as a way of classifying plants [51,104]; rather, that term refers to a combination of attributes of a crop that are contextual. Typically, the most important of these attributes are (1) high labor input required to make a useful product and (2) low market value. In a social environment where women's status and their agency in farming is low, responsibility for any farm product with this unattractive combination of attributes is relegated to women. In effect, "women's crop" does not refer to the plants but to the gender relations that legitimize unequal labor arrangements for use of a crop to the disadvantage of women [52,105]. Gender differences in varietal adoption decisions associated with crops that are nominally "women's crops" cannot, therefore, be interpreted as free choices exercised by women for a crop or trait of that crop.

Faced with disadvantageous product attributes, notably high labor input and low market value, women's trait preferences can be "adaptive" leading to self- subordinating choices, such as their preference for low-performing native potato varieties cited earlier [46]. A study of 658 farm households conducted in the Indonesia, Laos, Philippines and Vietnam found that where men and women had equivalent shares in the labor of rice cultivation, consensus about rice varietal trait preferences was high, but where male labor dominated the system there was less agreement and some women had different, self-subordinating preferences, leading to their favoring lower yielding rice varieties [63]. The important point here is that different trait preferences cannot be interpreted automatically as the product of meaningful choice, because the parameters of choice can be constrained by unequal gender relations that make self-subordinating choices rational, albeit disadvantageous to women.

Understanding varietal trait preferences as a function of gender relations helps to explain why gender differences in preferences are not fixed over time and can be quite plastic, depending on changes in gender roles and norms, the evolution of production, processing and market conditions for the crop in question. As conditions change, an erstwhile "women's crop" can become more commercialized or gender relations become more equitable, varietal preferences of men and women may converge around the traits required for successful market penetration [81,100,105].

In sum, what do gender differences in the adoption of improved plant varieties tell us about meaningful versus restricted choice in relation to women's empowerment? Women producers tend to adopt improved varieties at a lower rate than men: women producers sometimes express different preferences from men for varietal traits and these preferences affect their varietal adoption decisions. Differences between men and women producers in trait preferences often reflect underlying gender inequalities that constrain choice. Even as gender difference is only one facet of the intersecting social identities that influence choice of technology, our objective here is to analyze choice in relation to women's empowerment, so our focus is on gender. Improved varieties often do not incorporate traits that women producers value. On occasion, this oversight can be a "deal-breaker" as far as adoption goes: if women dislike the variety, they may convince men not to grow it. Although very few studies have examined the empirical relationship between gender-differentiated trait preferences and sex-disaggregated rates of adoption of improved varieties [21,22], the evidence suggests that women producers frequently find improved varieties ill-adapted to their needs and constraints, as illustrated with case studies in the next section [106–110]. This situation exemplifies restricted technology choice. As a result, when breeders make decisions about which varietal traits to select for in a new variety, they are determining whose preferences are going to be privileged or restricted. In the next section, we consider a framework for analyzing how breeders' decisions about which traits to prioritize can determine who gets a meaningful technology choice.

## 4. Gendered Parameters of Technology Choice in Development-Oriented Plant Breeding

In this section, we examine how development-oriented plant breeding processes construct technology choices for resource-poor men and women producers and how these can have a built-in gender bias. The case studies discussed here provide examples of breeding programs that have documented important changes in their priorities that were made to address gender issues and provide women producers with meaningful choice.

This analysis focuses on a pivotal decision in plant breeding: trait prioritization. Plant breeders' trait prioritization is a decision process in which the parameters of choice are being set for producers' future variety adoption decisions. Conventionally, this is a decision process in which users of improved varieties have little decision power, although their preferences as consumers may be consulted. Plant breeding programs generally aim to supply varieties to farmers with improvements of specific traits such as high yield, disease resistance or drought tolerance. To develop new plant varieties, breeding programs typically generate through crossing programs a large number of promising plant materials that are progressively reduced in successive cultivation cycles to select a few. In the private

sector, selection is guided by criteria provided by the marketing team responsible for identifying the product profile for a given customer segment that the business has decided to target. In the public sector and in developing countries where market analysis is less available, breeders tend to guide selection more independently. To guide selections, private sector and some public sector breeders first develop a product profile that describes the set of desirable traits of the new plant variety with target values for the key traits which can be feasibly attained (e.g., level of resistance to disease). This requires a process to evaluate, weigh and prioritize the individual plant traits under consideration for inclusion in the product profile. For practical reasons, the number of traits that can be included in any one profile and addressed by one breeding program is restricted [111], though this has been changing with genomics assisted breeding methods. The final product profile defines the set of traits that will be used to set breeding objectives throughout the selection process and, ultimately, the parameters of choice for producers' future adoption of the new variety.

Decisions about which traits to incorporate in a new plant variety, and which and how many product profiles should be included in a breeding program, are a demonstration of how social understandings, categories and schema are inscribed into choices made in the process of research [112]. Decisions about trait prioritization involve choices that draw on deep-laid, taken-for-granted normative schemas for the interpretation of gender about whose preferences are prioritized and who will benefit [45,84].

In the latter portion of this section, we present a simple framework for analyzing the gender balance of a choice. We then apply the framework to trait prioritization and illustrate how some breeders have rebalanced trait prioritization to provide women producers with meaningful choice, drawing on a number of plant breeding experiences. Meaningful choice is presented in this framework as a precondition for empowerment. Table 1 lays out schematically the gender balance of choice when an option, such as a varietal trait, is prioritized. Prioritizing a given varietal trait highly valued by men producers is compared with prioritizing one highly valued by women producers. For example, situation A (Column 1, Row 1 of Table 1) illustrates an option that is highly valued by both men and women. A variety with high yield can be valued positively by both men and women and represents an equal choice of a valued option. In contrast, situation B (found in Column 1, Row 2 of Table 1) illustrates the situation when an option is highly valued by men producers but is of negative value to women. For example, a grain that is high yielding could be positively valued by men producers who sell the extra grain. However, the variety is negatively valued by women because, being high yielding, it generates more unpaid work for women who do all the work of threshing. In this example, hypothetically, the women settle for lower yield if it means less work (i.e., they are unwilling to make the trade-off that increases their work burden). In situation B, a decision to prioritize the high yielding option tips the balance of power in the choice in favor of men. Thus, women's adoption choice is restricted, even if women and men producers have the same opportunity to choose this variety and to obtain seed of the new variety. However, situation B could be changed to situation C (Column 2, Row 1) if the option is a new variety that yields the same as the current variety in farmers' fields (and is thus "Indifferent" to men selling the grain) but is selected by breeders to be easier to thresh (also indifferent to men but highly valued by women). This constitutes a different balance of power in choice. Even more desirable would be shift of Situation C to situation A, in which women value the higher yield just as much as men because breeders have incorporated selection of traits that make the variety high yielding *and* even easier to thresh.

**Table 1.** Gender balance in choices valued by men and women.

| Prioritized Option Is | Highly Valued by Men | Indifferent for Men | Negatively Valued by Men |
|---|---|---|---|
| Highly valued by women | Gender equal choice of a valued option (A) | Gender unequal choice, in favor of women (C) | Gender unequal choice, in favor of women |
| Indifferent for women | Gender unequal choice, in favor of men (B) | Gender equal indifference | Gender unequal choice, in favor of women |
| Negatively valued by women | Gender unequal choice, in favor of men | Gender unequal choice, in favor of men | Gender equal choice of an unattractive option |

## 5. Case Studies

Using the framework for analyzing gender balance presented in Table 1, this section presents a series of cases where we illustrate the situation of restricted choice for women producers when traits they value are overlooked. The focus of this analysis is on providing examples of how individual choice and agency is constrained by structures beyond their control and how institutional objectives and priorities need to change for individual decision-makers to exercise agency through meaningful choice. We interpret this institutional change as a necessary precondition for empowerment. However, these cases were not designed to provide a detailed record of overall change in women's empowerment.

Case studies are used because there is a shortage of long-term panel studies using large-scale surveys (above 500 respondents) with representative samples of well-defined, sex-disaggregated customer segments that (a) identify a divergence between what breeding programs offer and what women growers and other end users demand, and (b) follow-up on what happens to adoption if more gender-responsive breeding objectives are set and more acceptable varieties released [22,101]. Many studies of trait preferences are not designed to analyze gender differences: this is one lesson from the analysis that follows. Table 2 below summarizes the gender balance of the original choice in these cases and Table 3 summarizes the shift in the gender balance towards meaningful choice that occurred as a result of changes in trait prioritization. Meaningful choice must involve the existence of choice and the perception of choice among options that have value to the actor and are available to him or her. Meaningful is not the same as empowered choice, but it is a precondition for empowerment to occur [1]. The analysis of cases demonstrates that a shift in the gender balance from restricted to meaningful choice is not automatically empowering. When changes are made in the structure of choice to include a new and meaningful option, this in itself does not guarantee individual empowerment because the opportunity to exert individual agency in technology choice continues to be structured in important ways by other decision-makers.

**Table 2.** Gender balance of the original choice in the case studies.

| Prioritized Option Is | Highly Valued by Men | Indifferent for Men | Negatively Valued by Men |
|---|---|---|---|
| Highly valued by women | Gender equal choice of a valued trait | Gender unequal choice, in favor of women | Gender unequal choice, in favor of women |
| Indifferent for women | Gender unequal choice, in favor of men | Gender equal indifference | Gender unequal choice, in favor of women |
| Negatively valued by women | Gender unequal choice, in favor of men | Gender unequal choice, in favor of men | Gender equal choice of an unattractive trait |
|  | • Hard shell groundnut (Malawi) | • Hard to peel cassava (Nigeria)<br>• Sorghum with poor yield on low phosphorus soils (Mali) | • Slow cooking beans (East Africa)<br>• High yielding, disease resistant matoke varieties with poor sensory quality traits (East Africa) |

**Table 3.** Gender balance in choice after changing trait priorities for the cases described.

| Prioritized Option Is | Highly Valued by Men | Indifferent for Men | Negatively Valued by Men |
|---|---|---|---|
| Highly valued by women | Gender equal choice of a valued trait<br><br>• Low P tolerant Sorghum (Mali)<br>• Faster cooking beans (East Africa)<br>• Softer shell groundnut (Malawi)<br>• Hybrid matooke with sensory quality traits (East Africa) | Gender unequal choice, in favor of women<br><br>• Easy to peel cassava (Nigeria) | Gender unequal choice, in favor of women |
| Indifferent for women | Gender unequal choice, in favor of men | Gender equal indifference | Gender unequal choice, in favor of women |
| Negatively valued by women | Gender unequal choice, in favor of men | Gender unequal choice, in favor of men | Gender equal choice of an unattractive trait |

*5.1. Low P Tolerant Sorghum in MALI*

5.1.1. Gender Balance in the Original Choice

At the beginning of the sorghum breeding process in Mali [106], adoption of newly bred sorghum (*Sorghum bicolor* (L.) Moench) varieties had been relatively low in Mali for a long time. Sorghum is generally considered a "men's crop" in Mali and West Africa and so breeders were prioritizing traits based on what they knew about men's varietal preferences and selection of breeding materials on land owned by men. This presented women producers with a restricted adoption choice because women were, in fact, growing sorghum. However, the improved sorghum varieties available were selected for relatively fertile soil conditions. Women's fields were often less fertile than the fields cultivated by men because they were not allowed access to manure and could not easily get fertilizer. As a result, the improved sorghum varieties did poorly on women's plots and few women benefited from them. Soil testing confirmed that soil phosphorus deficiency was a major constraint causing late flowering and lower yields, and that plots cultivated by women were, on average, well below the threshold for phosphorus deficiency in sorghum.

5.1.2. Gender Balance after Changing Trait Priorities

Research showed that adaptation to phosphorus-deficient soils would be a highly desirable trait from the perspective of women producers. In response, the sorghum breeding team implemented a series of changes in the procedures used to select and evaluate experimental sorghum varieties for early generation yield trials and design of farmer-managed variety trials. The sorghum program of the Institut d'Economie Rural and the International Crops Research Institute for the Semi-Arid Tropics (ICRISAT) in Mali undertook the challenge to breed varieties with better performance under phosphorus-limited conditions. As a consequence, the sorghum-breeding programs in Mali now grow all early generation material under low-phosphorus conditions in fields managed specifically for this purpose. Routine yield trials are now conducted under both high- and low-phosphorus conditions. Women farmers gained the option of a variety that would yield more than local varieties in their fields, because of its tolerance to low-phosphorus conditions. The availability of varieties that responded to women farmers' needs and constraints meant that women were no longer relegated by breeders, extension agents, their communities and families to the role of low adopters stymied by low productivity. With the reorientation of institutional objectives and priorities necessary for introducing meaningful choice, women's active participation in varietal evaluation increased and gave them more voice and agency in selection, seed multiplication and distribution.

*5.2. Groundnut in Malawi*

5.2.1. Gender Balance in the Original Choice

In Malawi and throughout East Africa, the groundnut (*Arachis hypogaea* L) breeding programs led by ICRISAT and the National Smallholder Farmers' Association of Malawi (NASFAM) were promoting two varieties [107]. One was groundnut rosette disease (GRD) resistant (one of the main diseases that impacts groundnuts and is a high priority in the groundnut breeding program) but with a hard shell. The other had a relatively softer shell but was not very tolerant to GRD. Shelling groundnut is very labor intensive, especially when the shell is hard to crack, and is customarily women's work in Malawi. It is particularly important that groundnuts are dried properly and stored in dry conditions. If the shell is very hard, women soak groundnut pods in water to soften the shell, then crack the shells by hand or mouth to access the nuts. The process of soaking hard-shell pods prior to shelling is very conducive for fungal growth, aflatoxin infestation and contamination of the nuts. This contamination was serious and may have contributed greatly to Malawi's loss of its share of the European market. Farm women, therefore, preferred groundnut varieties that had a softer shell. Within approximately 7 years (2003–2010), the softer shell variety reached high levels of adoption whereas the other, more GRD-resistant variety was adopted at very low rates. A survey conducted in 2010 in the two main groundnut growing regions of Uganda compared plots managed by men with plots managed by women. The study found gender differences in trait preferences including ease of hulling: women groundnut producers prioritized local varieties and obtained lower yields than men whether with local varieties (a gap of 44%) or improved (a gap of 63%) [68].

5.2.2. Gender Balance after Changing Trait Priorities

The groundnut breeding program realized that the hard-shell varietal trait was a serious obstacle to adoption and refocused the breeding objectives towards delivering a groundnut variety that is similar to the preferred variety in terms of shell quality but also resistant to GRD. Changing the gender balance in trait priorities was integral to prioritizing two groups of traits that respond to demand from different actors in the groundnut value chain: input traits that include adaptation to biotic stresses such as GRD, and output, end-user traits that include ease of shelling [113].

*5.3. Beans in East Africa*

5.3.1. Gender Balance in the Original Choice

Common bean (*Phaseolus vulgaris*) is an important food crop in eastern and southern Africa and traditionally a subsistence crop managed by women. Recently, common bean has become more of a commercial crop with men increasing their involvement in all facets of its production, marketing and consumption. Breeders actively consulted men and women producers about their preferred traits [108]. Cooking time was discussed in these consultations but seemed to be of secondary importance and breeders had not prioritized it. However, research using choice experiments discovered that men were just as interested as women in varieties with shorter cooking time, due to the high cost of firewood and charcoal used for boiling dry beans. In eastern Kenya, male respondents were more likely to value short cooking time than women, which reflected the scarcity and high cost of cooking fuel. Generally, results showed that there is a sizeable demand for a new bean variety with short cooking time.

5.3.2. Gender Balance after Changing Trait Priorities

The study raised awareness among breeders of the importance of the short cooking time trait. Breeders invested in screening for short cooking time during the early breeding stages and started to preselect for this trait before materials were taken for on-farm evaluation. Cooking time is now considered one of the criteria for selection of potential varieties for release. Women and men producers gained the option of a fast-cooking bean that is of importance to both. Breeding has to progress further towards release of fast-cooking

varieties before the wide-ranging effects of having a meaningful choice of variety can be assessed.

### 5.4. Cassava in Nigeria

5.4.1. Gender Balance in the Original Choice

Cassava (*Manihot esculenta* Crantz) is an important staple for smallholder farmers in Nigeria. A nationwide survey of cassava growing regions asked a representative sample of producers to rank cassava varietal traits in order of importance and revealed that women producers in all regions disliked cassava that was difficult to peel. Cassava breeding to that point had prioritized improving yield, disease resistance and starch content, and had not paid attention to the peelability of different varieties [109].

Women in Nigeria and elsewhere can often work long hours on their own account or in cottage industry, peeling cassava on a semi-industrial scale for processing into numerous food products in popular demand. When cassava is difficult to peel, women have to spend more time on peeling. They also have to discard some of the good flesh of the cassava tuber with the peel. Being hard to peel lowers the efficiency of women's labor but also lowers the crop's economic yield. A processing trait like this one is more important to women producers because women are much more involved than men in cassava processing. Men producers were aware that some varieties were difficult to peel but ranked this trait as less important than numerous other traits, such as yield.

5.4.2. Gender Balance after Changing Trait Priorities

Cassava breeding identified women producers and processors as a specific beneficiary target group. The national Cassava Monitoring Study found that one of the most important traits mentioned, especially by women, was "ease to peel," highlighting the importance of understanding traits related to cassava quality and processing, in tandem with a thorough gender analysis, to setting future targets for cassava breeding [114,115]. Breeding priorities changed to assess the feasibility of selecting for ease of peeling and other quality traits in the development of new varieties.

### 5.5. Matooke in Uganda

5.5.1. Gender Balance in the Original Choice

Matooke, a variety of plantain (*Musa acuminata*), is an important food crop in East Africa. Hybrid varieties released have superior yield but poor taste [110]. Men and women both disliked the hybrid varieties, but for different reasons. Men were concerned about traits that affect marketability, taste in particular. Women were more concerned about cooking quality: low heat retaining capacity made the matooke hybrids harden very fast when served and required a long period of cooking to soften. Longer cooking time and hard consistency when cooked were important to women because these traits involved more work and use of scarce fuel.

5.5.2. Gender Balance after Changing Trait Priorities

The breeding program made a change in the sequence of trait selection. New matooke hybrids are now first evaluated and selected for sensory quality traits over traits such as yield and pest and disease resistance that previously took precedence. By shifting its focus to women, breeding programs provided women the option of choosing new varieties that do not demand more of their labor.

Table 3 is intended to be compared with Table 2. It summarizes the shift in gender balance towards meaningful choice that occurred as a result of changes in trait prioritization in the cases discussed above. In each case, changes in trait prioritization were made by the breeding program to introduce a new option, creating a meaningful choice of variety by incorporating a trait valued by women.

These cases illustrate how the choice of producers, men or women, in their decision to adopt an improved crop variety is structured by the priorities set by plant breeders.

Because they can decide whether or not to select for plant traits valued differently by men and women, breeding programs have the "power to" determine the gender balance of meaningful choice, and whose choice is restricted. In two of the cases, the initial trait prioritization situation restricted women's choice and this had negative implications, as women producers chose a variety susceptible to disease because it was easier to harvest (groundnut in Malawi) or were excluded from having the option to plant higher yielding improved varieties (sorghum in Mali). In another case (cassava in Nigeria), an option advantageous to women was not included in the originally prioritized traits, which meant women's heavy workloads and wasteful practices in cassava peeling were protracted rather than alleviated. However, in response to gender analysis that provided information on these issues, program priorities changed to construct new meaningful choices.

The cases illustrate how a gender imbalance in choice for adopters of new varieties is built into the way the technology is designed. This design of a new variety, embodied in the traits prioritized, powerfully affects who benefits from the technology and how they benefit. Choice is either constrained or enabled by decisions about technology design taken in institutions outside the ambit of the individual agency of technology adopters. Changes in breeding priorities that facilitate meaningful choice illustrate the importance of including changes in institutional policy for understanding and implementing strategies to address empowerment and sustainable development.

## 6. Discussion and Conclusions

What can be concluded from the experiences in plant breeding of changing program priorities to construct new options for meaningful choice? There have been several efforts to open up trait prioritization to input from small holder farmers in low-income countries without explicit consideration of gender differences, principally through Participatory Plant Breeding (PPB), Participatory Varietal Selection (PVS) and mother-baby trials. Their experience demonstrates that long-term commitment to consulting farmers' opinions or involving farmers directly in selection can produce notable improvement in adoption rates and increases in the release of varieties highly appreciated by farmers, especially women farmers in Africa [101,116–119]. Although focused on actively recruiting input from farmers on their trait preferences, PVS and related approaches did not generate decision-support tools that explicitly required consideration of gender differences. As a result, development-oriented breeding, which has the goal of contributing to gender equity and empowerment of women, has a practical challenge in how to systematize relevant information about gender differences, in a way that breeders can factor it into their trait prioritization. One response to this need has been to develop and validate a set of decision-support tools—the G + Tools—to provide breeding programs with a way to assess the gender balance of choices at key decision points in the breeding process.

The empowerment of women is seen as crucial for making progress out of poverty, and development interventions frequently include women's empowerment as a program objective. Yet, established approaches to women's empowerment emphasize individual agency and choice at the expense of tackling the issue of *how* the decisions or choices in question are defined. The empowerment process in development cannot achieve its goals so long as it focuses exclusively or primarily on improving women's ability to make choices without changing the way institutions structure the parameters of choice.

We suggest that the analysis of gender balance in plant breeders' trait prioritization provides a useful model for an assessment of the structure of choice for the exercise of agency, that are also inherent in other types of program priorities. There is a growing call for attention to "gendered technology development" in modern biotechnology and gene-editing [120,121]. This analysis adds strength to that argument by illustrating how gender is implicated in mundane technology design, calls for a new framework for "socio-genomic" research that recognizes that technical decisions are contingent on social categories like gender in ways that are not self-evident [112]. We have shown with the case of plant breeding, how technical program decisions structure choice and determine who benefits.

We also show how ostensibly value-free, technical decisions are contingent on deep-seated, normative dispositions about gender that frame assumptions (for example, that farmers are men and sorghum is a "men's crop"), as well as blind spots about the relevance that the impact on work assigned to women (for example, cooking, threshing or peeling) has on farmers' acceptance of a new variety. These normative dispositions are part of the intellectual underpinnings of classical plant breeding. The classical approach overrides gendered farmer knowledge, traditionally rooted in stewardship of sustainable plant biodiversity, in favor of work with heritable traits to enhance yields, profitability and uniformity within a narrow genetic base. The choices between plant traits analyzed in the cases are themselves restricted by the objectives of classical breeding, which is only one paradigm among many.

Thus, whether meaningful choice is created for an individual, depends on his or her goals and values *and* the worldview and values of others who exercise power over the structure of choice. The cases show how, even in a situation where the individual has agency in choice of technology, that same choice depends on other actors beyond the individual's control. Use of gender analysis enables recognition of these normative dispositions and their consequences, leading to changed priorities and different outcomes. A balanced approach to agency and empowerment of women and ultimately sustainable development requires careful consideration of the elements of choice.

**Author Contributions:** Conceptualization, methodology and formal analysis, V.P. and J.A.A.; resources, G.T. and H.T.; data curation, J.A.A.; writing—original draft preparation, V.P. and J.A.A.; writing—review and editing, V.P., J.A.A., G.T. and H.T.; supervision, G.T.; funding acquisition, G.T. and V.P. All authors have read and agreed to the published version of the manuscript.

**Funding:** The research to prepare this study was undertaken as part of, and funded by, the CGIAR Research Programs on Roots, Tubers and Bananas (RTB), the CGIAR Gender and Agriculture Research Network, and the CGIAR Gender Platform, all supported by CGIAR Trust Fund contributors https://www.cgiar.org/funders/ (accessed on 23 February 2021).

**Institutional Review Board Statement:** Not applicable.

**Informed Consent Statement:** Not applicable.

**Data Availability Statement:** Not applicable.

**Acknowledgments:** We would like to thank the three anonymous reviewers who provided insightful comments and suggestions to strengthen the arguments in the paper. This paper was coordinated by the CGIAR Gender and Breeding Initiative, the CGIAR Research Program on Roots, Tubers and Bananas, the CGIAR Research Program on Policies, Institutions and Markets.

**Conflicts of Interest:** The authors declare no conflict of interest. The funders had no role in the design of the study; in the collection, analyses, or interpretation of data; in the writing of the manuscript, or in the decision to publish the results.

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
