# Peer review of "When Is Choice Empowering? Examining Gender Differences in Varietal Adoption through Case Studies from Sub-Saharan Africa"

_sustainability, doi:10.3390/su13073678_

Round 1
Reviewer 1 Report
Gender paper revue.
This paper is well-written, but some of the gender-jargon is dense. For a nonspecialist the terminology is adapted from other disciplines and hard to follow. The plant breeding example chosen to illustrate and justify female gender inequity in decision-making and workload is a rather trivial case by now. The discussion is naïve about plant breeding and crop variety adoption. There are many examples of bigender research on participatory plant breeding that help to advance the concept of reorganizing gender empowerment. Plant breeding programs have suffered from lack of innovative leadership, but are addressing the gender gap by new leadership and especially with surging interest by females to join the plant breeding profession. I am not a specialist in the subject of this paper, but I would be better informed by the paper if it had a broadened scope about gender issues. For example, the mechanization of rice production has relieved females from planting, cultivation, and harvest. What are women now doing to contribute household economies? Small enterprises have come into the realm of activity for women. That is good, but a labor gap has arisen in some areas such that males from other countries are imported to meet certain aspects of “modernized” rice production.
The paper presents a small number of crop commodity examples where survey methods have identified the female gender inequity and have been modified by adjustment of the role of women in decision-making about plant breeding goals. These are fine examples, but for a serious exposition about the problem and proper adjustments, I would prefer to see examples of survey data that supports problem identification, proposed solutions, and actual results of follow-up surveys.
The paper would be strengthened by more detailed exposition about the advantages and problems exposed in the cassava and matooke (plantain) examples. The famous Rwanda bean breeding selection work by women could be mentioned, even though the ultimate outcome was affected by civil strife.
The paper would also be strengthened if details of the female empowerment process in the example crops were explained, along with positive and negative outcomes. In my view, a positive outcome would be the evolution of bigender ‘ownership’ and management of the food production process at the farm level. Surely there is more than new crop variety adoption that plays; for example, harvest methods, safe product storage, availability of planting stocks, and household financial income, to name a few.
In summary, I am interested in the topic and expect a broad readership about this topic. The paper is long on complicated rhetoric but short of details of practice and vision for the future as may be realized by gender balance in choice, breeding, and utilization of several hundred crop species. Publication of the paper ‘as is’ would be a disservice to interested readers about the topic and to the special collection of papers being assembled.
Notes to editors:
I suggest that the authors be strongly encouraged to simplify but expand the scope of the paper. The editors of the special collection should see my comments and judge the relevance of my concerns.
Author Response
|
Comments from Reviewer |
Response to reviewer |
|
This paper is well-written, but some of the gender-jargon is dense. For a nonspecialist the terminology is adapted from other disciplines and hard to follow. |
Significant reduction of section 2 by reducing the discussion of power, agency/structuration and habitus (starting at line 162-165; 167-199) |
|
The plant breeding example chosen to illustrate and justify female gender inequity in decision-making and workload is a rather trivial case by now. |
|
|
The discussion is naïve about plant breeding and crop variety adoption. There are many examples of bigender research on participatory plant breeding that help to advance the concept of reorganizing gender empowerment. Plant breeding programs have suffered from lack of innovative leadership, but are addressing the gender gap by new leadership and especially with surging interest by females to join the plant breeding profession. |
We have included references to the success of participatory breeding and varietal selection with including feedback from farmers in variety selection and release and have clarified that these examples did not generate decision-support tools that explicitly required consideration of gender differences. See clarifications in lines 666-670 |
|
I am not a specialist in the subject of this paper, but I would be better informed by the paper if it had a broadened scope about gender issues. For example, the mechanization of rice production has relieved females from planting, cultivation, and harvest. What are women now doing to contribute household economies? Small enterprises have come into the realm of activity for women. That is good, but a labor gap has arisen in some areas such that males from other countries are imported to meet certain aspects of “modernized” rice production. |
We added reference to the participatory breeding and varietal selection success with including feedback from farmers in variety selection and release, and clarified that these examples have not consistently built gender into the approaches. (line 666-676)
|
|
The paper presents a small number of crop commodity examples where survey methods have identified the female gender inequity and have been modified by adjustment of the role of women in decision-making about plant breeding goals. These are fine examples, but for a serious exposition about the problem and proper adjustments, I would prefer to see examples of survey data that supports problem identification, proposed solutions, and actual results of follow-up surveys. |
Case studies are used because, there is a shortage of long-term panel studies using large-scale surveys (above 500 respondents) that characterize representative samples of well-defined, sex-disaggregated customer segments, and that (a) identify a divergence between what breeding programs offer and what women growers and other end users demand; and (b) follow-up on what happens to adoption if more gender-responsive breeding objectives are set and more acceptable varieties released. (lines 481-486) Many studies of trait preferences are not designed to analyze gender differences: this is one lesson from the analysis that follows. (Lines: 486-488) |
|
The paper would be strengthened by more detailed exposition about the advantages and problems exposed in the cassava and matooke (plantain) examples. |
Some minor clarifications were included on matooke (See line 619). However, the details of every specific crop and the dynamics are not subject of analysis, but only serve as examples to focus on how choices can differ.
|
|
The famous Rwanda bean breeding selection work by women could be mentioned, even though the ultimate outcome was affected by civil strife. |
Even though the ultimate outcome was affected by civil strife, we made specific reference to the success of participatory varietal selection involving women farmers in increasing the release and adoption of improved bean varieties not only in Rwanda but in Eastern Central and Southern Africa. We then point to the issue that these approaches have not consistently required gender analysis so that taking a participatory approach to breeding has not made sure that gender was taken into account in trait prioritization. As a result, there is no guarantee that varieties presented to women farmers for participatory evaluation have taken women’s trait preferences into account. This is the point we want to illustrate with plant breeding as an example: if you deconstruct the elements of technology choice, there are different sets of decisionmakers involved, some of whom have more “power to” determine the outcome than others. (line 666-676)
|
|
The paper would also be strengthened if details of the female empowerment process in the example crops were explained, along with positive and negative outcomes. In my view, a positive outcome would be the evolution of bigender ‘ownership’ and management of the food production process at the farm level. Surely there is more than new crop variety adoption that plays; for example, harvest methods, safe product storage, availability of planting stocks, and household financial income, to name a few. |
Our focus is on how the breeding priorities changed to create meaningful choice but there is not always data or documentation available to detail if or how this change produced other forms of empowerment. We have included references to the participatory breeding and varietal selection success with including feedback from farmers in varietal selection and release and clarified the gaps. (line 666-676)
|
|
In summary, I am interested in the topic and expect a broad readership about this topic. The paper is long on complicated rhetoric but short of details of practice and vision for the future as may be realized by gender balance in choice, breeding, and utilization of several hundred crop species. Publication of the paper ‘as is’ would be a disservice to interested readers about the topic and to the special collection of papers being assembled.
|
Section 2 has been simplified in discussion of power, agency/structuration and habitus (starting at line 162-165; 167-199). Details of practice are simple and basically used to describe a gap in empowerment theory. The understanding of this gap will aid the development of a new research agenda on gender that looks critically at how choices are structured to generate institutional innovations that address this gap. In (lines 77-82, 91-94, 108-123) the focus of the paper is expanded and clarified. |
Reviewer 2 Report
Dear Authors,
That was very good read and I cannot say anything negative about your manuscript!
Best of Luck!
Author Response
|
Comments from Reviewer |
Response to reviewer |
|
That was very good read and I cannot say anything negative about your manuscript |
Thank you for your feedback it is very encouraging. |
Reviewer 3 Report
The manuscript by Polar et al. reviewed various scenarios of impact of gender inclusion in the successful adoption of varieties of various crops like peanut, sorghum, common bean, cassava, and plantain developed in the context of African nations. This manuscript concludes that plant breeding undertaken in considerations of gender equity and women empowerment particularly towards making choices for varietal selection for their farm can lead to development of new varieties that can be adopted by producers of both gender and could ultimately benefit the society. This study provides an excellent conclusion guiding plant breeding programs in the targeted regions to consider the needs of both genders. Although the manuscript is very important in the African agriculture context, the following comments could help in improving its quality.
Major comments
Section 1 and 2 of the manuscript fit broadly with the context and is not necessary to understand the objective of the manuscript. These sections can be significantly reduced and aligned more towards the goal of the manuscript to emphasize the gender in variety prioritization.
There are numerous writing errors in the manuscript. Therefore, it needs significant correction to be fit for the journal at this stage.
The title is vague given that most of the examples are from Africa. Mentioning Africa in the title would be more appropriate to convey the message.
Specific comments
Line 8: Mention the fourth author as indicated in the addresses or delete the superscript 4.
Line 12: Explain agency
Line 86: Remove extra period
Line 90: Explain “plant breeding has become”
Line 382: Should it be “… to select a few”.
Table 1:
- The negative gain for women producers could be attributed due to higher production. It would be more informative to know if the return, although low, increases at all for women when the new variety is adopted.
- It is not necessary to color the tabs since the text is self-explanatory. Make changes in other tables: 2,3 if applicable.
Line 441. Change P to phosphorus
Line 480: Mention the full name of the disease. Include groundnut rosette, if applicable, as done in line 506.
Line 514: … had not given it a priority.
Line 520. Insert period at the end of the sentence.
Line 530: Insert botanical names for all crops to maintain consistency as Cassava (Manihot esculenta Crantz)
Line 555: Incomplete sentence. Check both sentences. Explain matooke because it is not a crop but a variety of plantain
Line 587: Incomplete sentence.
Table 4. Provide the source and background how it fits in the context of the G+ consumer profile prototype in the text of manuscript.
Author Response
|
Comments from Reviewer |
Response to reviewer |
|
Section 1 and 2 of the manuscript fit broadly with the context and is not necessary to understand the objective of the manuscript. These sections can be significantly reduced and aligned more towards the goal of the manuscript to emphasize the gender in variety prioritization |
There was a significant reduction of section 2 by shortening the discussion of power, agency/structuration and habitus (starting at line 162-165; 167-199) |
|
There are numerous writing errors in the manuscript. Therefore, it needs significant correction to be fit for the journal at this stage. |
We thank you for your keen eye and input. The manuscript has been proofread again for copy editing |
|
The title is vague given that most of the examples are from Africa. Mentioning Africa in the title would be more appropriate to convey the message. |
The title has been changed to add “sub-Saharan Africa” at the end of the title
|
|
Line 8: Mention the fourth author as indicated in the addresses or delete the superscript 4.
|
The superscript has been adjusted to reflect the fourth author.
|
|
Line 12: Explain agency |
The definition of agency has been added to the abstract (line 12) and the introduction (line 42) There is also a definition of agency previously included in line 48-49 and supported by citation (14) |
|
Line 86: Remove extra period
|
The extra period has been removed
|
|
Line 90: Explain “plant breeding has become”
|
This sentence has been restructured. Se changes in lines 108-113 |
|
Line 382: Should it be “… to select a few”.
|
Yes, change made as per suggestion. |
|
Table 1: · The negative gain for women producers could be attributed due to higher production. It would be more informative to know if the return, although low, increases at all for women when the new variety is adopted. |
The description in Table 1 is hypothetical, see clarification in (line 457). Analysis of return is one possible way of approaching the assessment of positive or negative value for a target group. This type of analysis is not common and there are no known studies of this type highlighting gender differences.
|
|
· Table 1: It is not necessary to color the tabs since the text is self-explanatory. Make changes in other tables: 2,3 if applicable. |
All tables have been reformatted to remove shading and replacing red text with bold black font |
|
Line 441. Change P to phosphorus
|
We believe this comment refers to several mentions of P in section 5.1.2. We have replaced the P with Phosphorous in all mentions of the topic.
|
|
Line 480: Mention the full name of the disease. Include groundnut rosette, if applicable, as done in line 506.
|
Mention of rosette disease has been replaced with Groundnut Rosette Disease (GRD) throughout.
|
|
Line 514: … had not given it a priority.
|
This sentence has been revised and changes to… had not prioritized it (See line 577 – 578)
|
|
Line 520. Insert period at the end of the sentence.
|
The period has been inserted
|
|
Line 530: Insert botanical names for all crops to maintain consistency as Cassava (Manihot esculenta Crantz)
|
Beans, groundnut, sorghum and matooke have all been named with botanical names (line 506-507, 540-541, 572, 620)
|
|
Line 555: Incomplete sentence. Check both sentences. Explain matooke because it is not a crop but a variety of plantain
|
Matoke as variety clarified, sentence revised to: Matooke, a variety of plantain (Musa acuminata), is an important food crop in East Africa. Hybrid varieties released have superior yield but poor taste. (See line 620-621)
|
|
Line 587: Incomplete sentence.
|
Sentence adjusted to: However, in response to gender analysis that provided information on these issues, program priorities changed to construct new meaningful choices. (line 653-654)
|
|
Table 4. Provide the source and background how it fits in the context of the G+ consumer profile prototype in the text of manuscript.
|
Reference [123] included
|
Round 2
Reviewer 1 Report
The authors have edited the paper to expand on some points and clarifications of other points. I am comfortable with the revisions. The paper will serve the specialists in this field as they strengthen their research methodologies. The nonspecialist in gender-sensitive research will be informed, also somewhat distracted by the technical jargon adopted in this field of study.
Author Response
|
Reviewer 1 |
Response |
|
The authors have edited the paper to expand on some points and clarifications of other points. I am comfortable with the revisions. The paper will serve the specialists in this field as they strengthen their research methodologies. The nonspecialist in gender-sensitive research will be informed, also somewhat distracted by the technical jargon adopted in this field of study. |
Thank you for your feedback during the first round of reviews. They have significantly contributed to clarify the objective of the paper and to target the message better. We are glad you are now conformable with the revisions. |
Reviewer 3 Report
The manuscripts by Polar et al. reviewed the case of gender differences in varietal adoption in Sub-Saharan Africa and question about choice empowering but the content of the paper does not align with the title. The paper mostly talks about the differences and men and women for varietal acceptance and its uses. In most of the sections, the topic is generalized and more references should be included. Either the topic should not be vague or the manuscript should be divided or reduced with focusing only in the idea of the topic. It is difficult to keep track of the message being conveyed in some sections. The manuscript still needs significant revision to meet the journal’s quality.
Major comments:
There are many sentences that are not worded correctly and sometimes the whole paragraph sounds vague and out of place.
The ideas are spread all over the places rather than describing them in their own section.
Author Response
|
Reviewer 3 |
Response |
|
The manuscripts by Polar et al. reviewed the case of gender differences in varietal adoption in Sub-Saharan Africa and question about choice empowering but the content of the paper does not align with the title. |
Thank you for the observation. Following the academic editor’s suggestion and your observation we have now changed the title to 'When is choice empowering? Examining gender differences in varietal adoption through case studies from sub-Saharan Africa' |
|
The paper mostly talks about the differences and men and women for varietal acceptance and its uses. In most of the sections, the topic is generalized and more references should be included. Either the topic should not be vague or the manuscript should be divided or reduced with focusing only in the idea of the topic. It is difficult to keep track of the message being conveyed in some sections. |
References included in lines 77, 399, 401
Manuscript was reduced to focus on the idea of the topic. Reductions in lines 368-370 Reductions in lines 635-645 Sections 6 and 7 combined and reduce |
|
The manuscript still needs significant revision to meet the journal’s quality. |
With the support of a proofreader some changes to sentence wording and format were made, including all the issues highlighted in comments in the revised file. |